# Electric Fields Enhance Ice Formation from Water Vapor by Decreasing the Nucleation Energy Barrier

Leandra P. Santos [1] , Douglas S. da Silva [2] , André Galembeck [3] and Fernando Galembeck [1,2,*]

1   Galembetech Consultores e Tecnologia Ltda., Campinas 13080-661, Brazil;
    leandrapereiradossantos@gmail.com
2   Department of Physical Chemistry, Institute of Chemistry, University of Campinas,
    Campinas 13083-970, Brazil; dssilva@unicamp.br
3   Department of Fundamental Chemistry, Federal University of Pernambuco, Recife 50740-560, Brazil;
    andre@ufpe.br
*   Correspondence: fernagal@unicamp.br

**Abstract:** Video images of ice formation from moist air under temperature and electric potential gradients reveal that ambient electricity enhances ice production rates while changing the habit of ice particles formed under low supersaturation. The crystals formed under an electric field are needles and dendrites instead of the isometric ice particles obtained within a Faraday cage. Both a non-classical mechanism and classical nucleation theory independently explain the observed mutual feedback between ice formation and its electrification. The elongated shapes result from electrostatic repulsion at the crystal surfaces, opposing the attractive intermolecular forces and thus lowering the ice-air interfacial tension. The video images allow for the estimation of ice particle dimensions, weight, and speed within the electric field. Feeding this data on standard equations from electrostatics shows that the ice surface charge density attains $0.62$–$1.25 \times 10^{-6}$ C·m$^{-2}$, corresponding to $73$–$147$ kV·m$^{-1}$ potential gradients, reaching the range measured within thunderstorms. The present findings contribute to a better understanding of natural and industrial processes involving water phase change by acknowledging the presence and effects of the pervasive electric fields in the ambient environment.

**Keywords:** ice nucleation; surface tension; classical nucleation theory; electric field; non-classical particle formation and growth; environmental electricity

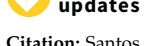



## 1. Introduction

Metastable conditions frequently arise due to slow water phase transition kinetics [1–4], such as liquid water supercooling or overheating [5] and water vapor supersaturation [6], which produce disastrous consequences. Some examples are the violent formation of thunderstorms, the rupture of water pipes due to sudden freezing, boiler explosions, and performance problems in vapor turbines [7]. Researchers and plant engineers thus provoke or intensify the nucleation of the thermodynamically stable phases to prevent problems derived from metastability in industrial and energy processes. However, metastability is a feature of ordinary glasses essential to modern life.

Following classical nucleation theory [8], Equation (1) [9] allows for the calculation of the homogeneous nucleation rate of a new phase:

$$R = N_s Z_j \, exp^{\left(\frac{-\Delta G^*_{homo}}{kT}\right)} \tag{1}$$

where $N_s$ is the number density of nuclei, $Z_j$ is the probability that a nucleus at the top of the energy barrier will grow further, forming a stable particle, and $\Delta G^*_{homo}$ is the energy barrier to the formation of nuclei, given by Equation (2):

$$\Delta G^*_{homo} = \frac{4}{3}\pi r^3 \Delta g_v + 4\pi r^2 \gamma \tag{2}$$

where $\Delta G^*_{homo}$ is the summation of two terms dependent on the particle radius, $r$. The first is the bulk contribution to Gibbs' energy change, below 0 °C, for ice formation, which is <0. The second is the surface contribution that reaches large positive values per mol when $r$ is in the nanometric size range.

Equation (3) gives the energy barrier for heterogeneous nucleation ($\Delta G^*_{hetero}$). It equals the energy barrier for homogenous nucleation ($\Delta G^*_{homo}$) multiplied by a function of $m$ and $x$, where $m$ is the interfacial interaction parameter $m$, and $x$ is the relative size of foreign particles [10].

$$\Delta G^*_{hetero} = \Delta G^*_{homo} f(m, x) \tag{3}$$

Following Equations (2) and (3), lowering the surface tension of the ice nuclei decreases the energy barrier of homo- and heteronucleation, thus increasing the ice nucleation rate and alleviating the problems caused by metastability.

However, phase separation follows non-classical mechanisms in many-component systems [11], such as the spinodal decomposition of metal alloys and polymer solutions, in crystal growth by particle attachment (CPA) [12,13], or by their assembly [14] in biomineralization [15,16]. Thus, non-classical phase separation and growth kinetics occur in a growing number of processes, but it does not yet appear in the current literature on ice crystallization from vapor [17,18].

Following theoretical predictions, the effect of electric fields on phase equilibria thermodynamics is negligible below $10^8$ V m$^{-1}$ [19]. However, the literature contains significant information on enhancing phase separation kinetics in electro-freezing and electro-crystallization [20–24], but the intervening mechanisms are disputed [25]. Intrinsic electric fields participate from colloidal particle growth processes, e.g., in the rod-dumbbell-sphere fractal growth of fluorapatite within a gel [26]. However, literature considering their presence is still scarce.

A recent report from this group showed the increase of ice formation under electric fields [27]. Now, the present paper explains this effect using classical nucleation theory. Additionally, it presents new experimental evidence of the concurrent participation of a non-classical particle formation mechanism. The two mechanisms operate by lowering or eliminating the energy barrier to ice nucleation.

## 2. Materials and Methods

### 2.1. Materials

Aluminum sheet used as electrode was bought from Mercadão do Ferro e da Chapa (Guarulhos, SP, Brazil). The graphite-coated paper sheet was prepared by coating Kraft Paper from Irani (Indaiatuba, SP, Brazil) using aqueous graphite-cellulose dispersions supplied by Galembetech (Campinas, SP, Brazil) based on the procedure described by Ferreira et al. [28,29]. Graphite was Grafine® 95100 from Nacional de Grafite (São Paulo, SP, Brazil) and cellulose was Microcel®-101 from Blanver (São Paulo, SP, Brazil). Liquid nitrogen was from White Martins (Campinas, SP, Brazil).

### 2.2. Equipment

A high-voltage power supply from Hypot DC Instrum (São Paulo, SP, Brazil) operating within 0 and −10 kV produced electrode bias. That instrument has voltage cut-off capability when the current rises to 100 μA for accident prevention. The Dewar was a GNL02 model, 2-L wide mouth, cylindrical double-walled pan, made from stainless steel and produced by Cryometal (Campinas, SP, Brazil).

The video recording instruments were an EOS Rebel T1i Canon camera with video capability and a Knup KP-8012 digital microscope (São Paulo SP, Brazil). Frame extraction used the Hitfilm Express 14 editing package running on an Aspire 5 Acer laptop computer. Frame identification followed the [hour:minute:second: frame number] pattern, where the frame number ranges from 0 to 23 (24 frames per second).

*2.3. Experimental Setup*

The experimental setup consisted of a stainless-steel Dewar bottle filled with liquid nitrogen overlaid with a horizontal aluminum plate measuring $30 \times 20 \times 0.3$ cm, held on two blocks of polyethylene foam, 5.5 cm above the Dewar. A conductive wire connected the Dewar to ground, and another connected the aluminum plate to a Hypot DC Instrum high-voltage power supply, creating a 1.8 kV.cm$^{-1}$ electric field. A schematic representation is in the Results section.

*2.4. Temperatures*

The thermometers were a Minolta Cyclop Compact 3 infrared thermometer and a k-type thermocouple connected to an E5CN Omron digital temperature controller. The thermocouple mounted on an x-y-z microscope positioner scanned the temperatures. A digital thermo-hygrometer (MTH-1380 Minipa, São Paulo, SP, Brazil) measured temperature and humidity in the room atmosphere.

**3. Theory and Calculations**

*3.1. Ice Formation Enhancement by Decreasing Its Surface Tension: Application of the Classical Nucleation Theory*

Equation (2) allows for the calculation of the effect of the ice-air interfacial tension on the energy barrier to ice nuclei formation. Its application requires data for $\Delta g_v\ (r)$ and $\gamma(r)$ when $r$ is within the nanometric domain. Those are thermodynamic quantities whose meaning is open to discussion in a system formed by water molecule clusters. However, assuming the validity of macroscopic values for $\Delta g_v\ (r)$ and $\gamma(r)$, Equation (2) yields the Gibbs energies for spherical ice particles relative to water vapor, shown in Table 1.

The calculation of $\Delta g_v\ (r)$ used Equation (4), following Job and Ruffer [30]:

$$\widetilde{\mu} = \widetilde{\mu}^{298} + \alpha(T - 298) \tag{4}$$

where $\widetilde{\mu}$ is the electrochemical potential of ice or water vapor at the temperature $T$, $\widetilde{\mu}^{298}$ is the respective chemical potential at 298 K under electric potential (V) equal to zero, and $\alpha$ is its temperature coefficient of the electrochemical potential.

The ice surface tension $\gamma(r)$ is equal to 0.090 N m$^{-1}$, neglecting its dependence on temperature and particle size [31–33].

The fourth column in Table 1 shows that forming small ice nuclei under $-5\ °C$ and $-15\ °C$ requires a highly positive Gibbs energy, explaining why ice is not homogeneously nucleated from vapor except at temperatures well below 0 °C. However, lowering the surface tension also decreases $\Delta G^*_{homo}$, bringing it within the same range as the RT product. Thermal fluctuations then allow ice nucleation to proceed.

Previous work from this group [34] showed how the surface tension of liquid water decreases by subjecting it to a positive or negative electric potential. When the potential approaches 10 kV, hanging water drops transform into jets, and sessile drops explode, emitting smaller charged droplets. It is then possible to say that the electrostatic repulsion of surface charges in electrified water counter the intermolecular forces and may even supersede them, producing a negative surface tension.

**Table 1.** Gibbs energy for the formation of small ice particles from vapor, at two temperatures below zero Celsius and considering four different values for the ice-air interfacial tension.

| Particle Radius (m) | Temperature (°C) | $\Delta g_v$ $(r)$ (J/mol) [a] | $\Delta G^*_{homo}$ (J/mol) $\gamma = 0.09$ N/m [b] | $\Delta G^*_{homo}$ (J/mol) $\gamma = 0.02$ N/m [c] | $\Delta G^*_{homo}$ (J/mol) $\gamma = 5 \times 10^{-3}$ N/m [c] | $\Delta G^*_{homo}$ (J/mol) $\gamma = 10^{-3}$ N/m [c] |
|---|---|---|---|---|---|---|
| $3 \times 10^{-10}$ | | | 17,965 | 3965 | 965 | 165 |
| $1 \times 10^{-9}$ | | | 5365 | 1165 | 265 | 25 |
| $3 \times 10^{-9}$ | | | 1765 | 365 | 65 | −15 |
| $1 \times 10^{-8}$ | −5 | −35 | 505 | 85 | −5 | −29 |
| $3 \times 10^{-8}$ | | | 145 | 5 | −25 | −33 |
| $1 \times 10^{-7}$ | | | 19 | −23 | −32 | −34 |
| $3 \times 10^{-7}$ | | | −17 | −31 | −34 | −35 |
| $1 \times 10^{-6}$ | | | −30 | −34 | −35 | −35 |
| $3 \times 10^{-10}$ | | | 17,715 | 3715 | 715 | −85 |
| $1 \times 10^{-9}$ | | | 5115 | 915 | 15 | −225 |
| $3 \times 10^{-9}$ | | | 1515 | 115 | −185 | −265 |
| $1 \times 10^{-8}$ | −15 | −285 | 255 | −165 | −255 | −279 |
| $3 \times 10^{-8}$ | | | −105 | −245 | −275 | −283 |
| $1 \times 10^{-7}$ | | | −231 | −273 | −282 | −284 |
| $3 \times 10^{-7}$ | | | −267 | −281 | −284 | −285 |
| $1 \times 10^{-6}$ | | | −280 | −284 | −285 | −285 |

[a] The value of $\Delta g_v$ $(r)$ is calculated using chemical potentials given by Equation (4). [b] Taken from Djikaev and Ruckenstein [35] but neglecting the temperature dependence of the surface tension. [c] The surface tensions in the three final columns are hypothetical values showing that negative $\Delta G^*_{homo}$ is achieved for nanometric particles but only for very small $\gamma$.

This section concludes that the energy barrier for water vapor condensation forming ice is vastly reduced or eliminated under an electric field, allowing rapid ice formation.

### 3.2. A Non-Classical Path for Ice Particle Formation Enhancement under an Electric Field

Water placed under a finite electric potential acquires excess charge opposite to the signal of the electric potential, following the electrochemical potential Equation (5):

$$\widetilde{\mu}_i = \widetilde{\mu}_i^o + \text{RT} \ln a_i + z\text{F}\psi \tag{5}$$

where $\widetilde{\mu}_i$ is the electrochemical potential of the ionic species $i$, $\widetilde{\mu}^o_i$ is the same but under standard conditions, $a_i$ is the activity of species $i$, $z$ is the number of unit charges per ion, $\psi$ is the electric potential in the domain that contains the ions $i$, R is the gas constant, T is the temperature, and F is the Faraday constant.

For instance, the equilibrium concentration of $H^+$ in pure water is $2.4 \times 10^{-4}$ mol·L$^{-1}$ when the water is under a 10 kV electric potential, which significantly departs from the $1 \times 10^{-7}$ mol·L$^{-1}$ standard concentration under zero electric potential. The charged water clusters that are thus formed depart from isometric shapes and develop an elongated body with a higher specific surface area to minimize electrostatic repulsion. Cluster elongation requires decreasing the number of H-bonds per water molecule compared to electroneutral bulk water. Energy for breaking H-bonds is available: the repulsion between two discrete charges in the opposite poles of a 0.5 nm diameter water cluster is in the 100–200 kJ·mol$^{-1}$ range, higher than the hydrogen bond enthalpy, 23.3 kJ·mol$^{-1}$. The resulting high specific surface area increases their surface energy and thus the tendency to accretion with neutral clusters.

This process proceeds through the successive and concurrent steps, schematically represented in Figure 1.

The proposition of this new path for ice particle formation derives from observing the sudden appearance of dim needles that become progressively brighter while their length and width do not change significantly with time. These features are presented in the experimental section and are akin to observations made during spinodal decomposition [36].

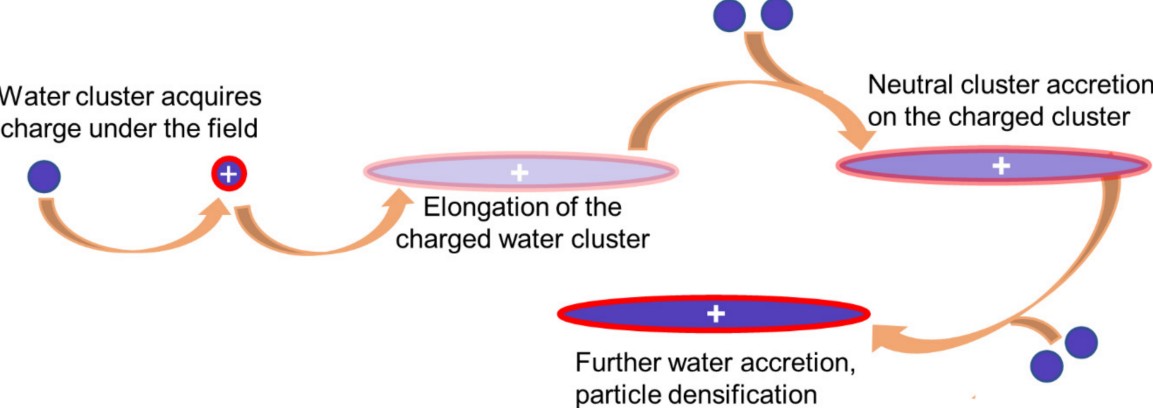

**Figure 1.** A Non-classical mechanism for ice particle formation from water vapor under the influence of an electric potential gradient.

## 4. Results

Records of most experimental observations in this paper are the video files accessible using the links listed in the Supplementary Material. Most of figures are video frames showing noteworthy events.

The experiments are initiated by filling the grounded Dewar bottle with liquid nitrogen and placing it within a Faraday cage under air (see schematic drawing and picture, Figure 2a,b). A turbid flat layer quickly appears floating ca. 3 mm above the bottle rim (shown in Figure 2, extracted from Video S1) and moving radially towards its borders to fall just after passing them. A thin layer of powdery ice covers the Dewar bottle rim within a few minutes. Figure 2c–f shows the formation of ice needles shortly after applying −10 kV to the upper aluminum plate. Needle formation is not expected under these conditions, considering that the elongated ice habits do not usually prevail [37] even in the limited temperature range where they appear. Assuming that the potential gradient is linear in the region between the bottle rim and the aluminum plate, the electric potential at the level of the condensation layer is ≈−545 V.

The temperature in the region just above and below the Dewar bottle rim is shown in Figure 3. The temperature in the condensation layer is between 5 and 9 °C, falling to −6 °C at the level of the bottle rim.

Ice dendrites grow on top of the needles, but the growth patterns of single needles and dendrites are different. Dim needles may appear abruptly levitating in the air (Figure 4, extracted from Video S2) or adjacent to a surface (Figure 5, see Video S3). In many cases, there is no sign of a needle in a frame, but it becomes noticeable in the next frame, just 0.04 s later (frame frequency is 24 s$^{-1}$), in areas bathed by aerosol from the condensation layer.

Figure 5 shows many needles measuring a few millimeters that appeared within a 1–2 s interval, with non-uniform visual contrast between different segments. This crystal formation pattern is different from the classical growth from a pre-existing nucleus (homogeneous or heterogeneous), and it has no precedent in vapor condensation studies. However, it is reminiscent of non-classical crystal growth or phase separation patterns in polymer membranes [38], metal alloys [39], glass [40], liquid-gas [41], and glass-gas [42] processes.

In Figure 6 (extracted from Video S4), the contrast-enhanced and false-color pictures show that ice grows much faster at the dendrite top than at its sides, or than needles or the packed ice beneath dendrites and needles. Table 2 presents ice growth rates measured from Figure 6.

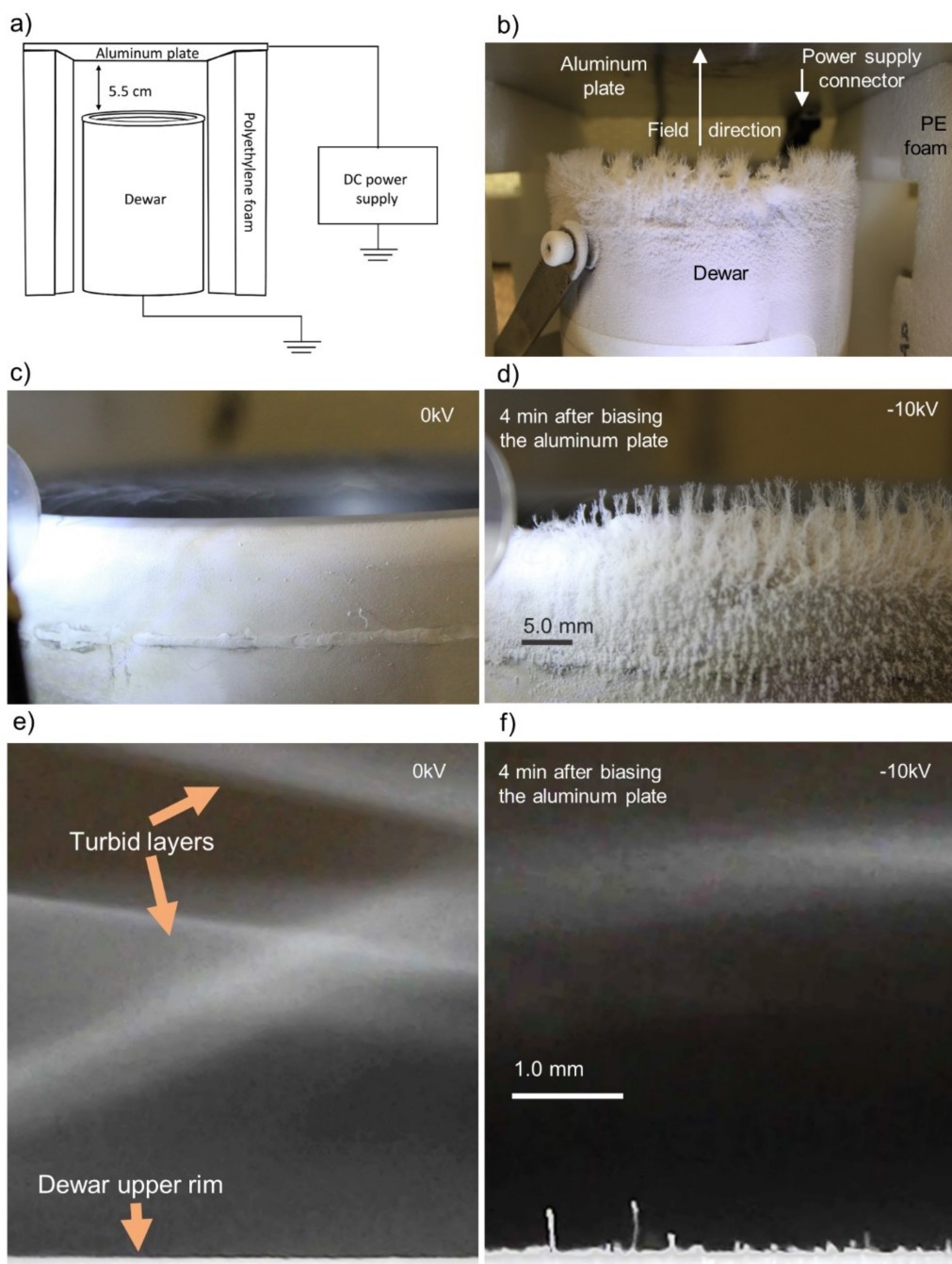

**Figure 2.** (**a**) Schematic drawing of the experimental arrangement. (**b**) Picture from the experimental set-up, showing the field directly above the Dewar rim. (**c**,**d**) Frames from the same area, showing needle formation on top of packed ice when the upper electrode (horizontal aluminum plate held 5.5 cm above the Dewar rim) is biased at −10 kV. Note the mobility of the turbid layer in the gas phase. (**e**,**f**) Frames from an area under intense lighting, showing needle formation on top of packed ice when the upper electrode (horizontal aluminum plate held 5.5 cm above the Dewar rim) is biased at −10 kV.

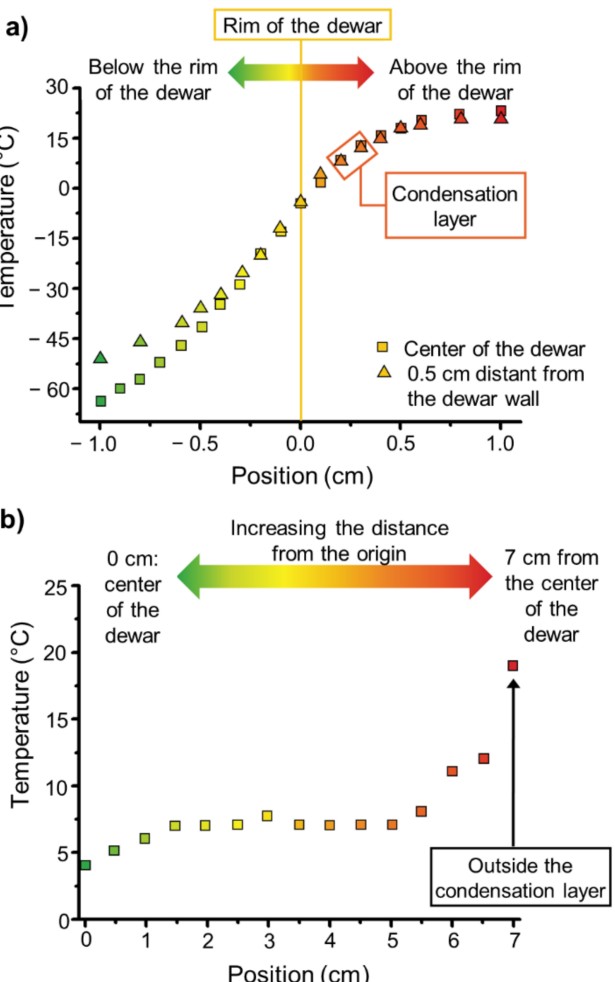

**Figure 3.** (**a**) Temperature variation along the vertical, measured with a thermocouple in the Dewar center and at 0.5 cm from its wall. The zero is at the rim height. (**b**) Radial temperature change in the condensation layer. The origin is the Dewar center.

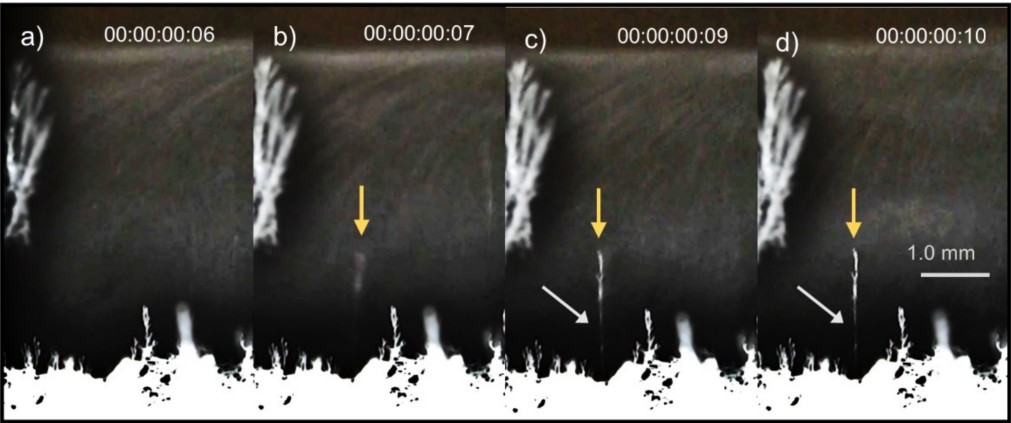

**Figure 4.** Consecutive contrast/brightness enhanced frames show the same area, where a floating ice needle (yellow arrows) appears in the lower part of each frame. There is no sign of a needle in frame (**a**) but it appears faintly in the following frame, Note the increase in needle contrast and its thickening, from (**b**–**d**) in the bottom, whereas its length remains nearly constant. The needle is not uniform along its length. Grey arrows point to a needle extension towards the lower surface, appearing only in the two frames to the right.

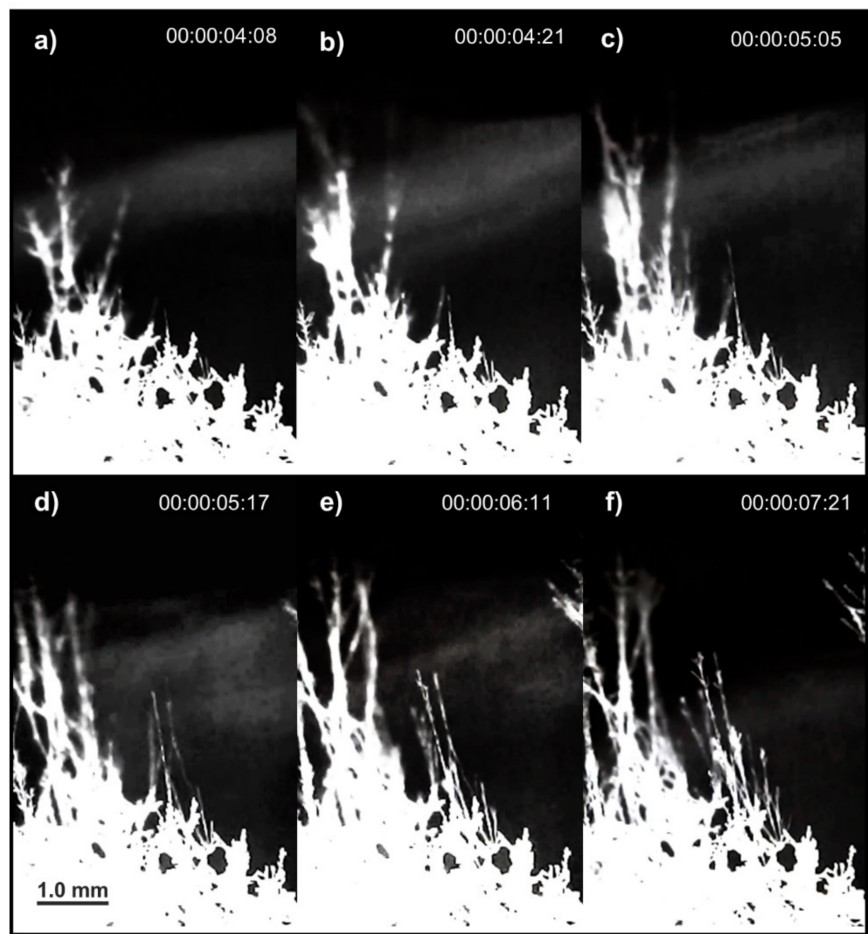

**Figure 5.** (**a–f**) Contrast-enhanced frames show the appearance of ice needles in the lower central part of each frame. Note that needles appear, and their contrast increases only in the presence of the supercooled water droplets appearing as a cloud in the background.

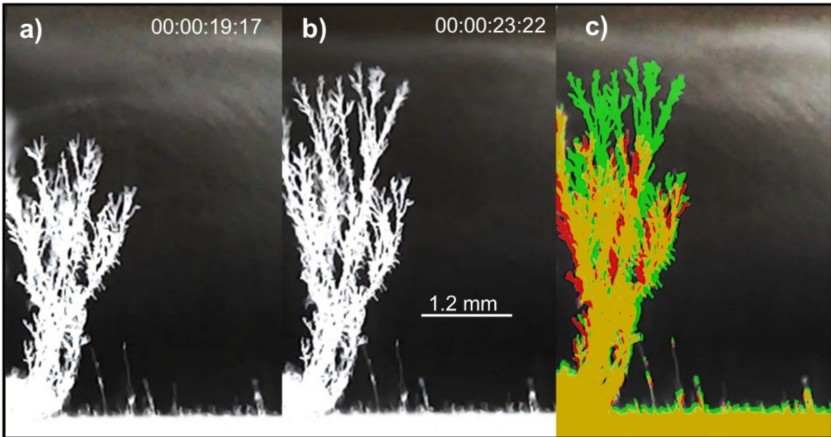

**Figure 6.** Dendrites and needle growth patterns. (**a,b**) Frames from the same area, taken a few seconds apart. (**c**) Combined (**a,b**) images with artificial colors: (**a**) red; (**b**) green. The ochre areas are where the colored pixel areas in (**a,b**) coincide. Note the dendrite displacement, upwards and to the right, where the cloud penetrates closer to the ice bed. Needles at the bottom of the pictures show only the non-uniform radial growth already presented in detail in Figures 4 and 5.

**Table 2.** Measured length or height of green sections of image components in Figure 6c. The time difference between frames b and a is 4.5 s.

| Image Component | Measured Length or Height/μm | Growth Rate/μm s$^{-1}$ |
|---|---|---|
| Top dendrite branches | 850 | 189 |
| | 810 | 180 |
| | 850 | 189 |
| | 890 | 200 |
| | 1280 | 284 |
| | 850 | 189 |
| | 770 | 171 |
| Lower dendrite branches | 150 | 33 |
| | 230 | 51 |
| | 120 | 27 |
| Needles | <40 | <9 |
| | <40 | <9 |
| | <40 | <9 |
| Lower ice layer | 40–80 | 9–18 |

Further crystal growth depends mainly on the thin cloud formed by water vapor condensation. Figure 6 shows dendrite top branches growing at a small lateral distance from idle needles that the condensation layer cannot reach.

Thus, the ice dendrites may grow as fast as 1 mm in 4−6 s, but their initial growth on top of the needles is much slower, as seen in Figure 7 (see Video S5).

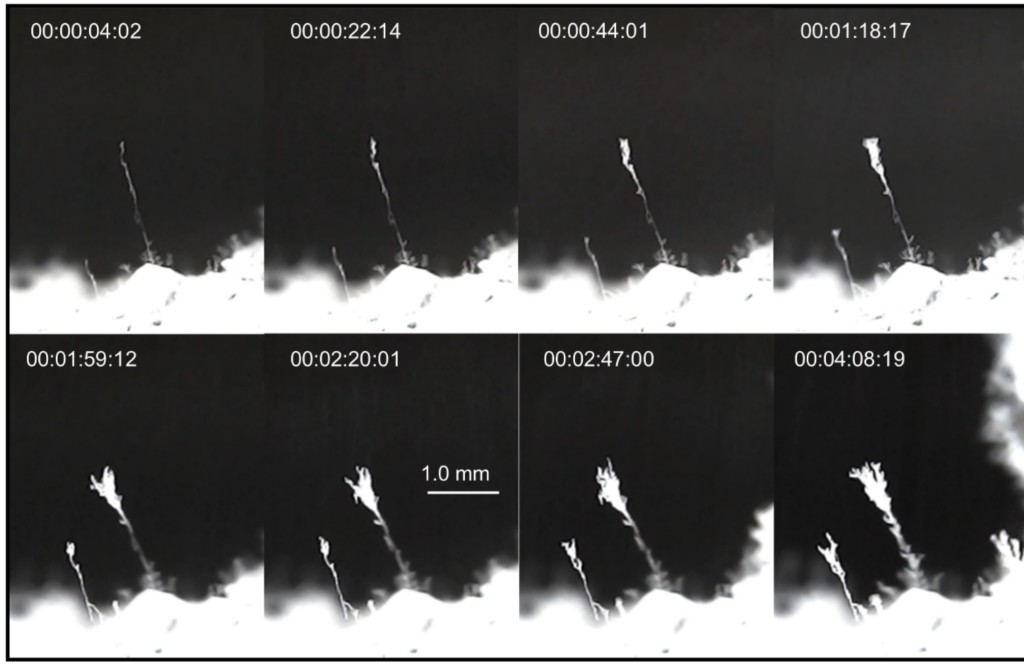

**Figure 7.** Frame series shows dendrite growth on the top and around isolated needles. Needles show only the non-uniform radial growth (presented in detail in Figures 4 and 5) for tens of seconds until their tops grow. The enlarged needle tops are the sites for the multiple branching in the later pictures. Short branches also bud along the lower sections of the needles, and their number increases, but not the lengths.

Site dependence of dendrite formation and growth rates results from the well-known effect of sharp tips on the electric potential distribution on any surface. Another interesting observation from these pictures is the dendrite branch rotation around the dendrite insertion

point. At the same time, the lower parts of the needles keep their positions, in agreement with an early report by Schaefer and Cheng [43].

Considering that ice stores charge forming electrets [44] and charge density determines electric potential, fast dendrite growth in some spots but not in others implies the formation of large potential gradients within the ice layer around the rim of the Dewar flask.

Electric fields change due to the water shielding ability. The direct measurement of potential gradients depends on the development of suitable miniature sensors currently being considered by the authors.

Both vertical and horizontal needles and dendrites fall downwards or curl following the electrode potential switching off, as shown in the two last video frames of the top line in Figure 8 (see Video S6). Reinstating the electrode voltage provokes upward motion of some dendrites and needles, while many others remain collapsed. The dendrites are thus jointed structures that are not rigid across their full extension.

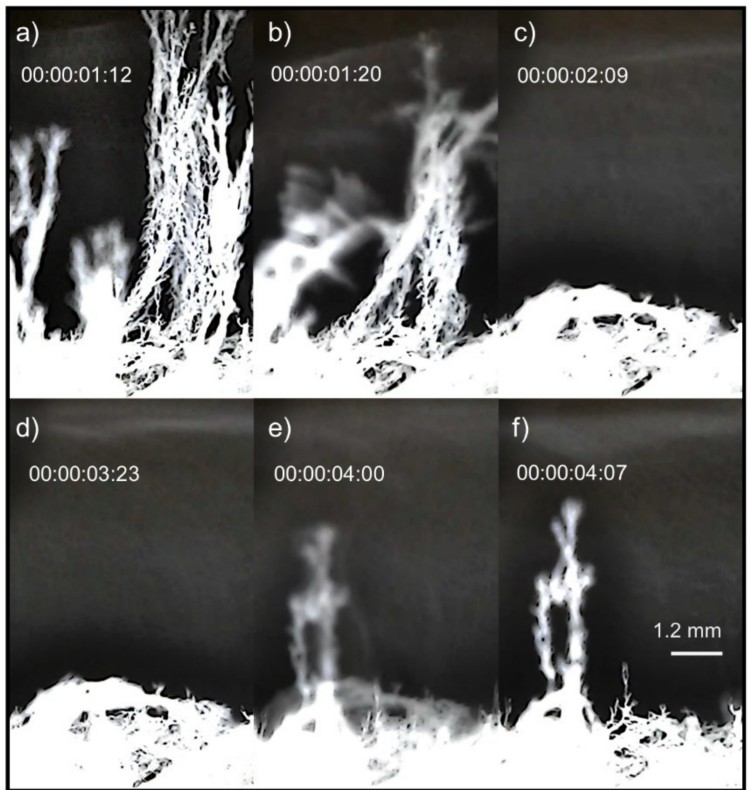

**Figure 8.** Sets of successive frames from the same area, showing the effect of switching the electrode voltage off and on. (**a**) Just before switching off; (**b**) just after switching off: dendrites are collapsing; (**c**) collapsed dendrites; (**d**) just before switching on again; (**e**) just after switching on, some dendrites and needles upraised under the field; (**f**) 400 ms after switching on.

The fallen dendrites form intricate networks that often show excellent morphological stability. However, these stable structures still change slowly under an external electric field (Figure 9, see Video S7).

Moreover, collapsed dendrites form packed ice under nil electrode voltage (shown in Figure 10 extracted from Video S8).

Figure 11 (see Video S9) shows dendrites and packed ice suddenly plucked from the supporting surfaces, flying towards the biased plate and thus disappearing from the image. That provides evidence that they also carry a net charge, analogous to previous water and ice electrification reports during condensation from water vapor or under an applied electric potential [45–48].

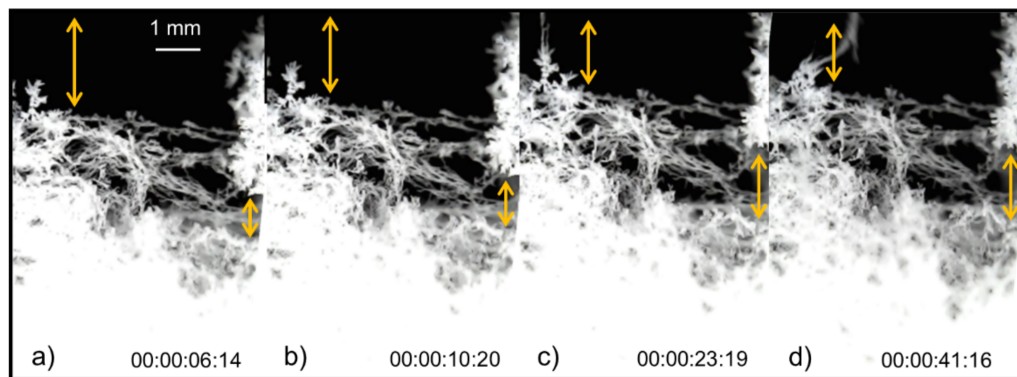

**Figure 9.** Frames from 6s14 to 41s16 show slight changes in the porous packed ice network triggered by the electric field. Switching the power supply 1 s before the frame in (**a**) provokes a slow displacement of the ice network from (**a**–**d**). The vertical yellow arrows indicate this change of position on the top-left and central-right parts of the frames. A network displacement of approximately 0.6 mm occurs where the ice appears more porous, whereas the structures in denser areas maintain their positions, with few changes.

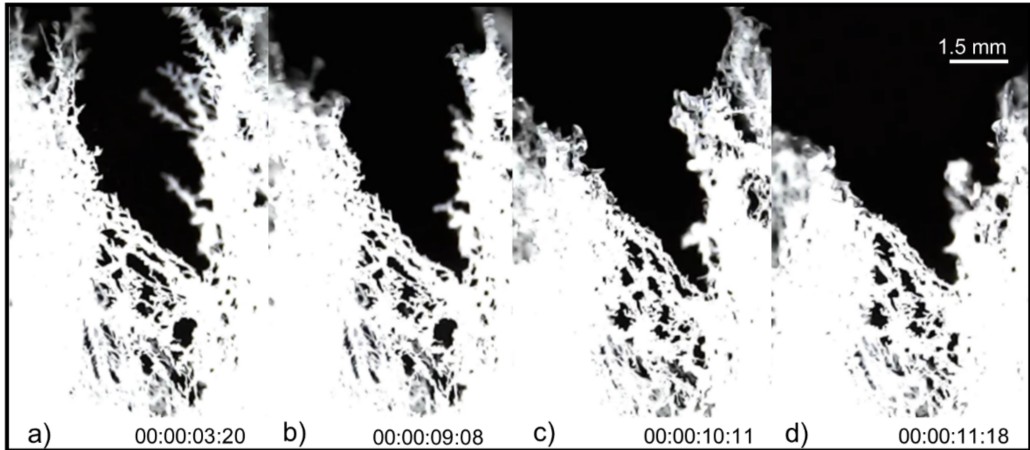

**Figure 10.** Pictures of frames from 3 s 20 to 11 s 18, showing dendrite transformation into packed ice in the absence of the applied field. Dendrites on the top of the porous network follow the potential gradient alignment in (**a**) but collapse under zero electric field, from (**b**–**d**).

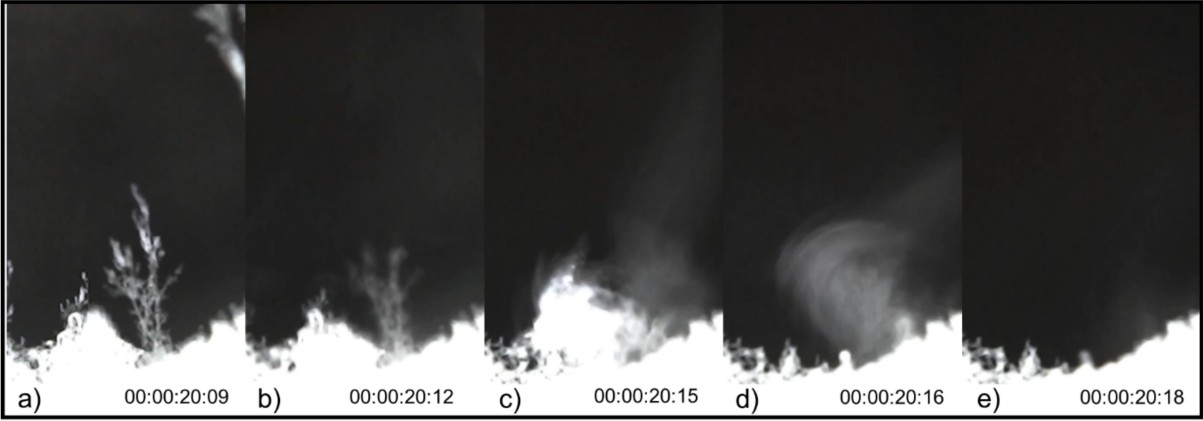

**Figure 11.** From left to right: frames (**a**) 20 s 09, (**b**) 20 s 12, (**c**) 20 s 15, (**d**) 20 s 16, (**e**) 20 s 18, showing the successive removal of dendrites and packed ice that are seen in (**a**) but not in (**e**).

The preceding paper [27] showed low-magnification images that allowed the estimation of the speed of flying ice needles, followed by the calculation of needle charge density

and thus of the electric field at the needle surface. These calculations followed standard equations from electrostatics. The magnified images shown in the present paper do not permit the observation of particle trajectories. However, they show detailed morphological information that allows the calculation of the weights of ice displaced upwards towards the biased electrode. Particle weight is the minimum electric force responsible for particles floating out of the ice bed. Knowing this force and the local electric field yields the overall particle charge. Combining that to the needle size and diameter yields the particle charge density and thus the electric field at the particle surface. A detailed example is in the Supplementary Information, together with data obtained from 14 other needles that jumped away from dendrites tips or the ice bed (see Supplementary Calculations and Equations section, Figure S1 and Table S1). The calculated charge surface densities range from 0.64 to $1.25 \times 10^{-6}$ C·m$^{-2}$, and the corresponding surface electric fields are within 75 and 147 kV·m$^{-1}$. The respective averages and standard deviations are $0.91 \pm 0.19 \times 10^{-6}$ C·m$^{-2}$ and $107 \pm 22$ kV·m$^{-1}$.

## 5. Discussion

Ambient electricity enhances ice formation rates and changes the shapes of ice particles formed from atmospheric vapor under low supersaturation (0 to $-6$ °C). Needles and dendrites are the prevalent ice habits formed under these conditions, compared to isometric ice particles formed within a Faraday cage. Two different mechanisms explain ice formation enhancement: one is non-classical, the other derives from classical nucleation theory. The coexistence of particle formation and growth mechanisms is not surprising within a system far from equilibrium and thus subject to large spatial and temporal fluctuations.

Both mechanisms converge, predicting the formation of ice with a high surface area displaying high charge density that accounts for the appearance of high electric potentials within the range observed in thunderclouds. Thus, ice formation and electrification mutually feed back until charge dissipation occurs by particle ejection or a Coulombic explosion takes place.

Surface tension lowering under an electric potential is well-known for water [34,49]. Still, there is not yet information on other systems. However, the same basic arguments used here probably have a broad application to other systems where particle nucleation and growth follow the classical theory. The present treatment of the effect of an electric field on ice nucleation from water vapor bears some resemblance to Tohmfor and Volmer [50], who added an electrical term to the Gibbs energy of growing particles to represent the effect of ions on their growth. However, the present treatment considers any intervening electric field beyond the electrostatic interactions intrinsic to the system.

The unexpected ice formation by a non-classical mechanism also depends on applying an electric field, producing electrified ice needles. In some pictures, the particle emerges from a surface, whereas in others, it grows adjacent to a surface but does not contact it. This phase growth proceeds by a gradual and non-uniform change in concentration within a given domain, evidenced by the increased contrast in the following pictures. That proceeds with little change in the outer needle dimensions.

There is currently no theory for this non-classical ice formation from vapor, and the literature does not yet show non-classical crystal formation and growth from the gas phase. However, future developments may proceed beyond the simple mechanism described in Figure 1, analogous to the spinodal decomposition, crystal growth by particle attachment, and other cases of non-classical phase separation and growth.

The enhanced ice formation and shape change due to non-zero electric potential may thus contribute to answering the open questions on secondary ice production in clouds [51]. It also addresses issues by Martin [52] in another review: *"What are the charging mechanisms to obtain $10^5$ V m$^{-1}$ in a cloud?"* and *"A possible important feedback is the alteration of ice nucleation kinetics by the presence of strong electric fields or by electric charges induced on ice ... "*.

## 6. Conclusions

Ambient electricity enhances ice formation rates and changes the shapes of ice particles formed from atmospheric vapor under low supersaturation (0 to $-6$ °C). Needles and dendrites are the prevalent ice habits formed under these conditions, rather than isometric ice particles formed within a Faraday cage. Two different mechanisms explain ice formation enhancement: one is non-classical, while the other derives from classical nucleation theory. Both predict mutual feedback between ice formation and electrification verified experimentally by detecting electrified particles whose surface charge density and electric field reach $0.64 \times 10^{-6}$ C·m$^{-2}$ and 147 kV·m$^{-1}$, respectively.

The present findings may contribute to a better understanding of environmental, chemical, and materials sciences and process technologies where phase changes are routine.

**Supplementary Materials:** The following supporting information can be downloaded at: https://www.mdpi.com/article/10.3390/colloids6010013/s1, Video S1: Needle and dendrite formation and change using the set-up shown in Figure 2a. Image acquisition was done using a digital video microscope; Video S2: Sudden appearance of ice needles floating in the air; Video S3: Sudden appearance of thin ice needles adjacent to a surface; Video S4: Vertical and lateral growth of dendrites exposed to cooled water vapor; Video S5: Dendrite growth on the top and around isolated needles; Video S6: Collapse of needles and dendrites in the absence of an electric field, when the aluminum plate is grounded; Video S7: Small changes in the porous packed ice network triggered by the electric field; Video S8: The dendrites collapse forming packed ice, under nil electrode voltage; Video S9: Removal of dendrites and packed ice; Supplementary Calculations and Equations: Estimates of needle charge, based on the minimum force required for ice extraction from the substrate; Figure S1: Needle extraction from a dendrite. The yellow ellipse in (a) encloses a needle (height h = 0.001 m and diameter 2r = 0.0001 m), that flew from a dendrite top, and is no longer seen in (b); Table S1: Height, radius, surface charge density, and electric field of charged needles extracted from the ice surface.

**Author Contributions:** Conceptualization, all authors; data curation, L.P.S. and F.G.; formal analysis, L.P.S., F.G. and A.G.; investigation, L.P.S., F.G. and A.G.; methodology, L.P.S., F.G. and A.G.; project administration, F.G.; resources, F.G.; software, D.S.d.S.; supervision, L.P.S. and F.G.; validation, D.S.d.S.; visualization, L.P.S., F.G. and A.G.; writing—original draft, and writing—review and editing, all authors. All authors have read and agreed to the published version of the manuscript.

**Funding:** This work is supported by the INCT program of the Brazilian agencies MCTIC/CNPq (465452/2014-0), CAPES and FAPESP (2014/50906-9), and by FAPESP PIPE project (2018/00834-2). LPS thanks FAPESP (2019/04565-9) for a research fellowship.

**Data Availability Statement:** The data presented in this study are available on request from the corresponding author.

**Acknowledgments:** The authors thank James P. Rydock and Earl R. Williams for their critical comments.

**Conflicts of Interest:** The authors declare no conflict of interest.

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
