# Peer review of "Electric Fields Enhance Ice Formation from Water Vapor by Decreasing the Nucleation Energy Barrier"

_colloids, doi:10.3390/colloids6010013_

Round 1

Reviewer 1 Report

The manuscript “Electric fields enhance ice formation…” by Santos et al deals with an interesting phenomenon and describes and interesting effect: The formation of ice crystals stimulated by electric fields. Therefore I think that the results are potentially publishable. However, in the present form the manuscript does not look like being finished. More like preliminary notes. For example, the first sentence (Dewar rim, as shown in Figure 2) does not seem to be a full sentence. Where is figure 1? I am missing a good description of the experimental setup. Therefore I recommend major revision so that at the ends a conclusive story is described. Then one may decside if the manuscript is worth publishing. A further suggestion is to include references with resepct to the effect of electric fields on the condensation of water, e.g. G.Z. Han, Q.X., J.Phys. Chem. C 124, 2020, 1820.

Author Response

A.: The Reviewers received damaged paper files. The Authors submitted a Word file that was damaged during its insertion in the journal template and we apologize for not having verified the result of the conversion. We are now submitting a file that is already within the template, showing all the expected sections for a full paper within a proper format.

The Han paper mentioned by Reviewer 1 is the new reference 17, lines 56-57. The corresponding text in the Introduction is: “Following theoretical predictions, the effect of electric fields on phase equilibria thermodynamics is negligible below 108 V m-1 [17]. However, the literature contains significant information on the enhancement of phase separation kinetics…”

Reviewer 2 Report

The manuscript claims the electric fields exhibit enhancement in ice formation from water vapor by decreasing the nucleation energy barrier. But authors are requested to explain or clarify the following arguments:

  1. Where is the description of figure 1 and Figure 1?
  2. Why are there two same sets of picture in the figure 4?
  3. The authors should explain in detail about the influence of electric fields on the ice growth rate?
  4. The authors should show the schematic diagram of experimental device?
  5. Why does the “cloud” appear in images shown in Figure 5?
  6. The authors should check the language very carefully as there are so many mistakes.

Author Response

A.: The Reviewers received damaged paper files. The Authors submitted a Word file that was damaged during its insertion in the journal template and we apologize for not having verified the result of the conversion. We are now submitting a file that is already within the template, showing all the expected sections for a full paper within a proper format.

Answers to the specific Reviewer 2 questions are beneath:

  1. Where is the description of figure 1 and Figure 1?

A.: They are at the end of Section 3.2, page 4, lines 160-163.

  1. Why are there two same sets of picture in the figure 4?

A.: The upper row shows the unedited frames for the sake of authenticity, while the lower row shows edited pictures, to facilitate the observation of the needle appearance and growth.

  1. The authors should explain in detail about the influence of electric fields on the ice growth rate?

A.: This is now explained in Section 3, pages 3-4. Two concurrent mechanisms operate to enhance ice growth rate: following the Classical Theory, the effect of the electric field lowering the interfacial tension of ice nuclei produces the decrease or elimination of the energy barrier to the formation of nuclei at the critical size. Another mechanism involves the deformation of small water or ice droplets accompanied by the accretion of other ice or water particles.

  1. The authors should show the schematic diagram of experimental device?

A.: Yes, this is now presented in Figure S1 of the Supplementary Material.

  1. Why does the “cloud” appear in images shown in Figure 5?

A.: The cloud is formed by condensed water vapor and it appears in most video frames. It cannot be observed only when the imaged area is very close to the Dewar rim and the surrounding areas contain tall dendrites.

  1. The authors should check the language very carefully as there are so many mistakes.

A.: We apologize for the truncated file received by the Reviewers. The present file was revised carefully.

Reviewer 3 Report

This article reported experimental results about ice formation under electric field. The authors reported multiple interesting observations, which could be potentially helpful for the understanding of ice formation in the artificial and natural environment. However, the manuscript is largely unmatured and needs to be significantly rewritten so that it can be understood by the audience. Detailed comments are given below.

  1. There is no introduction to the manuscript. The authors started the “Introduction” section with their results, which ignored the introduction of previous literature. There is no “Results” section in the paper.
  2. The authors offered a lot of figures in the manuscript but only described their experimental observations without fully explaining them from fundamental physics. One example is that the charge density is one critical parameter mentioned in the abstract and conclusions. However, there is very little discussion in the main text and there are no details on the calculation of the charge density range provided in the “Conclusion” section.
  3. More details need to be included about the classical and non-classical pathways used to descript the ice formation. The authors only offered qualitative discussion in the “Discussion” section. On line 173, the author wrote “There is not a theory for this Non-classical path”. I suggest the author to read the review article on this topic and re-evaluate their statement: “Crystallization by particle attachment in synthetic, biogenic, and geologic environments,” Science, 349, 6247 (2015).
  4. Reference styles are mixed and misleading. For example, on line 160, the author wrote “[15,xiii]”, which is confusing.
  5. I would suggest the authors avoid using Roman numbers for reference indexing. For example, on line 102, the author wrote “electretsviii”, which is hard to understand.
  6. The manuscript needs to be proofread carefully. Even the first sentence in the manuscript “Dewar rim, as shown in Figure 2,” is not grammarly complete. Also, there is no Figure 1 in the manuscript.
  7. The supplementary movie has a size of 13 GB, which is too large for readers to download. Please compress the movie to a reasonable size (~25 MB) and it should be uploaded into the journal’s official website, instead of sharing using a personal google drive link.

Author Response

A.: The Reviewers received damaged paper files. The Authors submitted a Word file that was damaged during its insertion in the journal template and we apologize for not having verified the result of the conversion. We are now submitting a file that is already within the template, showing all the expected sections for a full paper within a proper format.

Answers to specific questions are beneath:

This article reported experimental results about ice formation under electric field. The authors reported multiple interesting observations, which could be potentially helpful for the understanding of ice formation in the artificial and natural environment. However, the manuscript is largely unmatured and needs to be significantly rewritten so that it can be understood by the audience. Detailed comments are given below.

  1. There is no introduction to the manuscript. The authors started the “Introduction” section with their results, which ignored the introduction of previous literature. There is no “Results” section in the paper.
  2. Reference styles are mixed and misleading. For example, on line 160, the author wrote “[15,xiii]”, which is confusing.
  3. I would suggest the authors avoid using Roman numbers for reference indexing. For example, on line 102, the author wrote “electretsviii”, which is hard to understand.
  4. The manuscript needs to be proofread carefully. Even the first sentence in the manuscript “Dewar rim, as shown in Figure 2,” is not grammarly complete. Also, there is no Figure 1 in the manuscript.

A.: All these serious flaws were due to file truncation. They are now eliminated.

  1. The authors offered a lot of figures in the manuscript but only described their experimental observations without fully explaining them from fundamental physics. One example is that the charge density is one critical parameter mentioned in the abstract and conclusions. However, there is very little discussion in the main text and there are no details on the calculation of the charge density range provided in the “Conclusion” section.

A.: This is now addressed in the “Results” section of the revised manuscript, page 11, lines 293-308. A detailed description of the calculation of charge density is now presented in the Supplementary Information, showing data from fourteen withdrawn needles.

  1. More details need to be included about the classical and non-classical pathways used to descript the ice formation. The authors only offered qualitative discussion in the “Discussion” section. On line 173, the author wrote “There is not a theory for this Non-classical path”. I suggest the author to read the review article on this topic and re-evaluate their statement: “Crystallization by particle attachment in synthetic, biogenic, and geologic environments,” Science, 349, 6247 (2015).

A.: The new files show a comprehensive introduction with references to the literature on related subjects. The Science paper mentioned by the Reviewer is the new Reference 10 (page 2, line 52) on CPA, crystallization by particle attachment. We also included references on oriented attachment, nanoparticle self-assembly, and other bioinspired mechanisms and mesocrystals. All these different mechanisms are treated together with spinodal decomposition as Non-classical mechanisms for particle formation and growth. However, they still do not include electric field effects.

  1. The supplementary movie has a size of 13 GB, which is too large for readers to download. Please compress the movie to a reasonable size (~25 MB) and it should be uploaded into the journal’s official website, instead of sharing using a personal google drive link.

A.: Unfortunately, file compression produces a loss of resolution that would decrease the documental value of the video. For this reason, we opted for a video subdivision, showing only the sections containing the frames presented in the Figures of this paper. The resulting short videos are now included in the Supplementary Information.

Round 2

Reviewer 1 Report

The manuscript “Electric fields enhance ice formation…” by Santos et al deals with an interesting phenomenon and describes and interesting effect: The formation of ice crystals stimulated by electric fields. The manuscript has been improved significantly! It can now been published.

Here are some minor suggestions:

Isn’t in the first sentence abstract an “of” missing? “Video-recording of ice formation…“

  1. 2 l. 57: Ref 16 is missing.

Materials and methods: Sometimes is written in present, sometimes in past tense. I recommend to use past tense consistently.

Starting table 1 with 0.1 nm radius does not make much sense. That is a single molecule. I suggest to start at 0.3 nm or larger. In addition, it would be good to give a reason for the choice of the surface tensions.

Please explain the equation in l. 129, µ=µ25+aDt. What are the symbols? Where does the time-dependence come from? Why should the surface tension decrease?

  1. 156: Please explain the sentence “… the equilibrium concentration of H+ in pure water is 2.4 x 10-4 molL-1…”. Under which conditions?

I still think that a schematic of the experimental setup would help.

Where is figure caption 2?

In figure 4-7 the number of subfigures may be reduced.

Beginning of discussion: The categorization of ice formation enhancement does not seem to be very instructive to me. Under “non-classical” I would understand quantum mechanical effects.

Author Response

Reviewer 1

Comments and Suggestions for Authors

The manuscript “Electric fields enhance ice formation…” by Santos et al deals with an interesting phenomenon and describes and interesting effect: The formation of ice crystals stimulated by electric fields. The manuscript has been improved significantly! It can now been published.

Here are some minor suggestions:

Isn’t in the first sentence abstract an “of” missing? “Video-recording of ice formation…“

A.: This is now changed to “Video images of ice formation…”

 1.2  l.57: Ref 16 is missing.

A.: This is now corrected. Previous line 57. [15,15] is now changed to [17,18] in line 64 of the .docx file or line 69 of the .pdf file sent by the authors.

Materials and methods: Sometimes is written in present, sometimes in past tense. I recommend to use past tense consistently.

A.: This section is now consistently written in the past tense.

Starting table 1 with 0.1 nm radius does not make much sense. That is a single molecule. I suggest to start at 0.3 nm or larger. In addition, it would be good to give a reason for the choice of the surface tensions.

A.: The Reviewer suggestion is now implemented in Table 1, starting at 0.3 nm. The reason for the surface tensions in Table 1 is now given in footnote c).

Please explain the equation in l. 129, µ=µ25+aDt. What are the symbols? Where does the time-dependence come from? Why should the surface tension decrease?

A.: The Equation in l.129 is now rewritten as = (equation 4 of the new version of the article) where is  the electrochemical potential of ice or water vapor at the given temperature,  is the respective chemical potential at the temperature T and a is its temperature coefficient of the electrochemical potential. This information is now included in lines 124-128 of the .docx file or 133-138 of the .pdf file sent by authors.

Surface tension decreases due to electrostatic repulsion of surface charges, opposite to the attractive hydrogen bonding and other intermolecular forces that are ultimately responsible for surface tension. This is the subject of reference 34.

156: Please explain the sentence “… the equilibrium concentration of H+ in pure water is 2.4 x 10-4 molL-1…”. Under which conditions?

A.: This phrase is now corrected to “… the equilibrium concentration of H+ in pure water is 2.4 x 10-4 mol.L-1 when the water is under a 10 kV electric potential…”. See lines 157-158 of the .docx file or 169-170 of the .pdf file sent by the authors.

I still think that a schematic of the experimental setup would help.

A.: The schematic was shifted from the Figure S1 of the Supplementary Information to the main text. It is now shown in Figure 2a.

Where is figure caption 2?

A: In the .docx and .pdf files sent by the authors, the caption of Figure 2 is just beneath the Figure. In the combined .pdf file prepared by MDPI it was shifted to page 7, lines 193-196. For this reason, Reviewers 1 and 2 could not find it.

In figure 4-7 the number of subfigures may be reduced.

A.: The number of subfigures is now reduced, in Figures 4-7 and 9. Further reduction would eliminate essential information

Beginning of discussion: The categorization of ice formation enhancement does not seem to be very instructive to me. Under “non-classical” I would understand quantum mechanical effects.

A.: In the context of phase separation phenomena, “classical” refers to the phase separation resulting from particle nucleation and growth that was pioneered by Volmer and Weber (Reference 8). “Non-classical” refers to all the different phenomena and models that appeared since the spinodal decomposition was introduced by Cahn and Hilliard (Reference 11) in their classical study of physical metallurgy, followed by its application to polymer solutions. Other important mechanisms emerged more recently, and these are represented by References 12-18.

Reviewer 2 Report

I suggest to accept the paper as the authors have answered all the questions accordingly. And the manuscript looks goog enough to be published.

Author Response

The authors thank Reviewer 2 for the contribution to improve this paper.

Reviewer 3 Report

This reviewer appreciates the authors’ efforts to significantly improve the quality of this manuscript. However, I still find this manuscript has key flaws in its experimental design and explanation. Detailed comments are given below:

  1. Section 5 of the manuscript “Discussion” is unclear to me. Which part of the experimental results indicates the nucleation barrier has been lowered due to the existence of an electric field? To my understanding, this was only proposed in the theory section, while there is no experimental evidence to support the theory. For example, the nucleation could also be explained from a local concentration increase near the dewar, instead of the decrease of the nucleation barrier.
  2. The ice is growing on the surface of the dewar, which suggests this is not homogenous nucleation. However, in the theory section, the equations used by the authors are developed for homogenous nucleation, ignoring the existence of the substrate. The authors need to explain this.
  3. What’s the electric field distribution near the dewar? The alignment near the dewar needs to be illustrated as it will determine the strength and orientation of the ice.
  4. The voltage used in the experiments seems to be randomly chosen. It is suggested to study the dependence of nucleation phenomena at different voltages and compare with theory predictions. For example, how the nucleation rate and growth speeds depend on voltages.

Author Response

Reviewer 3

This reviewer appreciates the authors’ efforts to significantly improve the quality of this manuscript. However, I still find this manuscript has key flaws in its experimental design and explanation. Detailed comments are given below:

Section 5 of the manuscript “Discussion” is unclear to me. Which part of the experimental results indicates the nucleation barrier has been lowered due to the existence of an electric field? To my understanding, this was only proposed in the theory section, while there is no experimental evidence to support the theory. For example, the nucleation could also be explained from a local concentration increase near the dewar, instead of the decrease of the nucleation barrier.

A.: Figure 2 now shows macro pictures of the dewar top taken before and after switching the electric field, showing the enhancement of ice formation and the change in ice habit, after turning on the field. This takes place under temperatures close to 0oC and thus under low supersaturation, following the temperature profiles in Figure 3. Thus, switching on the electric field eliminates the metastability in ice formation. Since metastability is caused by slow nucleation that is in turn due to the nucleation energy barrier, the electric field acts by decreasing the energy barrier.

The local concentration increase near the Dewar does not explain the observed phenomena, since needles and dendrite do not appear in the control images acquired under 0 V, shown in Figure 2c and 2e. 

The ice is growing on the surface of the dewar, which suggests this is not homogenous nucleation. However, in the theory section, the equations used by the authors are developed for homogenous nucleation, ignoring the existence of the substrate. The authors need to explain this.

A.: Figures 4, 5 and other sections of videos S2 and S3 clearly show the appearance of needles suspended in the air, or formed close to the surface but later attaching to them. Please note that nanometer-sized nuclei cannot be currently observed by any imaging technique. The authors now included Equation 3 for heterogeneous nucleation, . Thus, any factor affecting  also affects

What’s the electric field distribution near the dewar? The alignment near the dewar needs to be illustrated as it will determine the strength and orientation of the ice.

A.: The direction of the field in the region where ice is formed is represented in the picture in Figure 2b. Thus, the needles are aligned along the field. This is evidenced also by the observed change in needle orientation when the field is switched on and off.

The voltage used in the experiments seems to be randomly chosen. It is suggested to study the dependence of nucleation phenomena at different voltages and compare with theory predictions. For example, how the nucleation rate and growth speeds depend on voltages.

A.: The voltage was chosen following the observations made during preliminary experiments, when the applied voltage was gradually increased, allowing the observation of the remarkable change in ice formation shown in Figure 2. We then chose a round number for the voltage.

The authors thank the Reviewer suggestion for future work. However, this requires a more sensitive way to detect ice formation and we are now examining some alternatives. New results could appear in six months.
